# Characterization of Technosols for Urban Agriculture

Borja Ferrández-Gómez [1] , Juana Dolores Jordá [2,*] , Antonio Sánchez-Sánchez [1] and Mar Cerdán [1]

[1] Department Biochemistry, Molecular Biology, Soil Science and Agricultural Chemistry, University of Alicante, 03690 Alicante, Spain; borja.ferrandez@ua.es (B.F.-G.); antonio.sanchez@ua.es (A.S.-S.); mar.cerdan@ua.es (M.C.)

[2] Institute for Environmental Studies, Ramon Margalef, University of Alicante, 03690 Alicante, Spain

*   Correspondence: juana.jorda@ua.es

**Abstract:** Soil characterization is essential for planning activities in urban areas in order to detect potential risks and understand the possible impacts derived from those activities. Nine soils located in Alicante (southeast of Spain) developed over construction debris were studied. Soil characteristics including mineralogy, elemental composition and metal availability were analyzed in two consecutive years, 2019 and 2020. These soils were similar to forest soils in the same area, with no evidence of asbestos clays or excess harmful elements. However, the use of DTPA extraction revealed high levels of Mn and Zn in some soils. Organic carbon and metals extracted with DTPA differed in 2019 and 2020, but no relationship between metal-DTPA and organic carbon content was observed. In general, organic matter content was higher in 2019, and elements extracted with DTPA were lower. The above-average rainfall in 2019 could have led to the washing away of dissolved materials and fine soil particles, decreasing elemental availability on the one hand, while promoting the development of natural vegetation, increasing soil organic matter, and immobilizing elements in living organisms on the other hand. The fact that the metal mobility varies depending on weather and soil characteristics is important when planning. Despite the demonstrated advantages of increasing urban green areas from an environmental and social point of view, we should not forget the materials on which urban soils are developed. Therefore, it is essential to establish annual plans for monitoring variations in the availability of heavy metals. This is of the most relevance when the plants are for human consumption. It is therefore also necessary to control the vegetables that grow on these soils and, in the event of possible problems, use the soil for gardening.

**Keywords:** asbestos; climate; heavy metals; organic matter; peri-urban agriculture; pollution; technosols

## 1. Introduction

In recent years, peri-urban agriculture and urban gardens have gained popularity for various purposes. Urban gardens improve the physical and social activities of certain groups, such as the elderly and the unemployed, or simply are a hobby for the populations of cities. Moreover, they contribute to the creation of greener cities, encourage the use of fresh, local products or even improve food safety [1–3]. These can also be used to inform citizens about issues related to biodiversity, recycling or circular economy [4]. Community gardens and other green infrastructures promote urban biodiversity and ecosystem services, mitigating the risks derived from the loss of natural habitats for local plant and animal populations [5].

Motivations for implementing urban agriculture are different according to the economic level of each country. In developing countries, food security and employment are priorities, while in developed countries, health and education issues are usually the main objectives [6].

Regardless of its use, the soils used for agricultural production in urban areas are far from being natural soils and may have suffered different anthropogenic impacts over

several decades. This local pollution must be taken into account before planning agricultural activities in these locations [7]. Soils often develop on the remains of rubble from old buildings or construction material dumps located on the outskirts of cities. These remains include bricks, concrete, Cu and Pb pipes and the rotten remains of metallic containers, which may contain heavy metals whose mobilization conditions are unknown. Concrete, for instance, is rich in Pb, Mn and Zn, although these metals are hardly lixiviated [8]. Lead is the main contaminant in urban soils with background values as high as 1000 mg/kg [9]. Lead is found in soils associated with organic matter, phosphates and iron oxides, although it is not clear whether or not the addition of these fertilizers can reduce the bioavailability of Pb. Brown et al. [9] concluded that soil treatments with these amendments considerably reduce Pb bioavailability and bioaccessibility. Cai et al. [10] also reported low bioavailability of Ba, Pb, Cu and Zn in urban soils, despite their presence in fine particles, although soil properties can modify bioavailability. These same authors reported opposite results [11] when they attempted to reduce the bioaccessibility of Pb and As by amending soils with organic matter, phosphate and iron oxide. After a five-year interaction, they found no reduction in the bioaccessibility of both elements. In arid and semi-arid regions, soil salinity and alkalinity impose serious restrictions on plant growth; salinity limits water uptake by plants and reduces germination, disperses clays and favors soil loss. (Solangi et al., 2019) [12]. Asbestos-containing products, roof sheets, pipes, tiles and pollutants [13] are also included in these potential hazards in urban agricultural areas. However, the occurrence of asbestos in urban agriculture has not yet been considered [14]. Six minerals are regulated as asbestos, one included in the serpentine group (chrysotile), and the other five are amphiboles (actinolite, tremolite, anthophyllite, crocidolite and amosite). All of them, except for crocidolite, contain Mg in their composition. Whether or not they can be considered asbestos, and thus pose health risks, depends on their form of crystallization [15]. In Spain, the use and commercialization of asbestos was prohibited in 2001 [16]. However, many buildings constructed before this ban still contain asbestos. Agricultural practices such as tilling can generate dust and asbestos dispersion that are responsible for severe health issues [17]. Soil pollutants can enter the food chain through edible vegetables, producing severe risks to humans [18].

Soil characteristics such as organic matter content, pH or metal competition affect the mobility and persistence of contaminants in soil [19,20]. The presence of lime favors heavy metal retention and limits their desorption [21], making it challenging to establish uniform safe metal levels based on general substance concentrations. This is especially difficult for heavy metals since they form part of soils naturally. For the establishment of reference values, several criteria have been taken into account, among them the use of the land (industrial, urban and other uses) and the routes of exposure. Urban use criteria also consider skin contact, while other uses consider the consumption of vegetables grown in these soils [22].

Soil characterization is essential for planning activities in urban areas in order to identify potential risks and assess the possible impacts derived from those activities. The aim of this paper is to analyze the suitability of nine Technosols for use in urban gardens for recreational purposes, including the study of the possibilities of infrared spectroscopy to detect the presence of asbestos in Technosols developed on dumping debris. Our hypothesis is that soils developed on construction debris may contain contaminating elements that could render them unfit for the production of edible vegetables and fruits. We applied simple analytical techniques to understand to what extent this hypothesis was true and how soil and environmental factors may affect soil pollution.

## 2. Materials and Methods

Nine Technosols from the peri-urban area of the cities of Alicante and San Vicente del Raspeig in southeastern Spain) were sampled, down to a depth of upper 20 cm. This is a developing area, where prior studies must be carried out in order to improve planning. The soils were collected to cover the whole study area (Figure 1). The majority of these

Technosols were shallow soils that had developed on construction debris dumps. In some cases, remains of pruning had been deposited for several years, while in others, these construction remains had been covered with soil, after leveling the area.

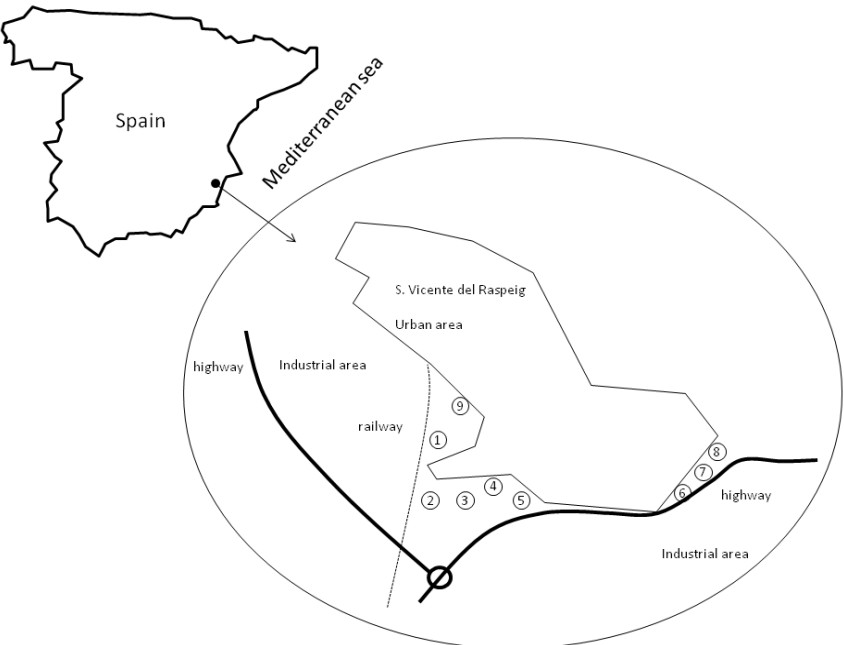

**Figure 1.** Map of the area showing the sampling area. The circles with a number inside indicate the sampling points.

There are two industrial areas surrounding the residential zone. The nearest buildings have been abandoned and the industrial area is moving further and further away from the population. Between these two industrial zones and in a more or less straight line is where the studied soils are located (Figure 1).

*2.1. Soil Characterization*

All measurements were made in triplicate, except for calcimetry, which was carried out in quadruplicate. Values of pH were determined using the saturated paste method [23], a 50 mL beaker was filled with 2 mm sieved soil and osmotized water was slowly added until saturation. The pH value was measured in this paste with a pH electrode. Calcimetry was used for lime concentration [24], where lime content was calculated by measuring the volume of $CO_2$ released when treating the soil with HCl diluted in water (1:1). Electrical conductivity (EC) was measured in soil water extracts. For this purpose, soil (20 g) was mixed with 100 mL of osmotized water in a plastic tube fitted with a sealed cap. It was stirred for 30 min, followed by centrifugation and filtration. The EC of the filtrate was then measured. Cation exchange capacity (CEC) was calculated by measuring cation displacement by $Na^+$. A total of 5 g of soil and 33 mL of sodium acetate 1 M were placed in a sealed plastic tube, stirred for 5 min and centrifuged. After discharging the supernatant, the operation was repeated twice. Then, the soil sample was washed three times with ethanol to remove all salt remains that could be present. Finally, the $Na^+$ retained in the soil was displaced with ammonium acetate and measured by Inductively Coupled Plasma Optical Emission Spectroscopy (ICP-OES, Perkin Elmer, Seer Green, Beaconsfield, UK, Optima 7300 DV). For the determination of organic carbon (OC), the Walkley–Black method was used [25]. Organic carbon was oxidized with potassium dichromate ($K_2Cr_2O_7$) in an acidic medium. The excess of dichromate was titrated using Mohr's salt ($Fe(NH_4)_2(SO_4)_2.6H_2O$) with diphenylamine as an indicator.

### 2.2. Soil Mineralogy

Soil mineralogy was investigated by Fourier Transform Infrared (FTIR, JASCO FTIR, Oklahoma City, OK, USA, 4700 spectrometer equipped with an Attenuated Total Reflectance (ATR) Specac Golden Gate accessory with monolithic diamond prism). Given the high carbonate content in these soils, the samples were treated with hydrochloric acid (5 g of soil + HCl 12 M until bubbling stops) in order to remove lime and improve clay signals. Following this treatment, the soils were repeatedly washed with osmotized water to remove any resulting calcium chloride and dried at 60 °C. Non-HCl-treated soil samples were also washed with osmotized water and dried at 60 °C in order to compare the FTIR spectra. Mineral samples of the University of Alicante collection and the RRUFF™ online database [26] were used as references. The spectra and the second derivative of the spectra were used to detect mineral characteristic bands.

### 2.3. Elemental Analysis

Heavy metals were measured by X-ray Fluorescence (FRX, sequential X-ray spectrometer (PHILIPS MAGIX PRO; Amsterdam, The Netherlands)) [27]. In order to calculate the percentages of heavy metals in soil, lanthanum oxide (9 g of soil + 0.1 g of $La_2O_3$) was added after confirming the absence of La in all soils. Since XRF provides information on the relative amounts of each element with respect to La and we knew the percentage of La added, we were able to estimate the percentage of each element in the soil [28]. Mercury content was measured directly in soil samples by using a mercury analyzer (Milestone Model DMA 80). In addition, to quantify heavy metal mobility, soil samples were stirred with the chelating agent DTPA (diethylenetriaminepentaacetic acid) [29]. The resulting extracts were analyzed by ICP-OES and ICP-MS) (Perkin Elmer, Optima 7300 DV and Agilent, mod 7700x, respectively).

### 2.4. Statistics

For each measurement, the mean values and standard deviation were calculated. Partial correlations between the different parameters were also calculated in order to verify the possible influence of different factors on soil pollution. All calculations were performed using Microsoft Excel™ (Microsoft Office LTSC Profesional Plus 2021).

### 3. Results

#### 3.1. Soil Mineralogy and Physicochemical Characterization

The analysis of soil mineralogy using FTIR resulted in the predominance of calcite and clays, such as illite and kaolinite (Figure 2). The presence of limestone is clearly evident in the FTIR spectra, by the bands at 1400 and 873 $cm^{-1}$. The band at 711 $cm^{-1}$ is characteristic of calcite. Notably, no spectra showed bands of other carbonates such as dolomite, which is also very common in the region.

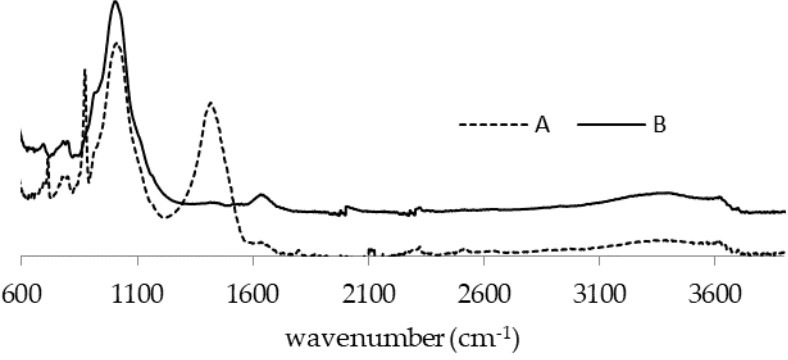

**Figure 2.** FTIR spectra of soil 6**A** Soil washed with osmotized water; **B**-soil treated with HCl to remove lime. The absence of signals at 1400, 873 and 711 $cm^{-1}$ is clearly observed after the treatment with HCl.

The analysis of lime content by calcimetry indicated that more than 55% of the soil samples was limestone, reaching 70% in some soils. Consequently, the pH values of these soils were high (Table 1). Interestingly, despite calcite being a mineral of moderate solubility, soil salinity was low.

**Table 1.** General physicochemical characteristics of the soils.

| Soil | pH | %Lime | EC dS/cm | CEC cmol·kg$^{-1}$ | %OC 2019 | %OC 2020 |
|---|---|---|---|---|---|---|
| 1 | 7.6 ± 0.1 ± 0.1 | 60 ± 2 | 0.18 ± 0.04 | 1.32 ± 0.05 | 1.3 ± 0.2 | 0.63 ± 0.02 |
| 2 | 8.1 ± 0.2 ± 0.2 | 70 ± 5 | 0.38 ± 0.04 | 0.7 ± 0.5 | 0.9 ± 0.04 | 0.54 ± 0.011 |
| 3 | 7.8 ± 0.1 | 59 ± 1 | 0.48 ± 0.04 | 2.4 ± 0.6 | 1.1 ± 0.1 | 0.82 ± 0.02 |
| 4 | 8.17 ± 0.05 | 68 ± 1 | 0.16 ± 0.05 | 1.6 ± 0.1 | 0.7 ± 0.03 | 3.9 ± 0.2 |
| 5 | 7.9 ± 0.1 | 58 ± 1 | 0.38 ± 0.02 | 4.1 ± 0.8 | 2.04 ± 0.04 | 1.10 ± 0.03 |
| 6 | 8.1 ± 0.1 | 56 ± 1 | 0.11 ± 0.03 | 9.5 ± 0.9 | 3.0 ± 0.1 | 2.51 ± 0.02 |
| 7 | 8.0 ± 0.1 | 57 ± 2 | 0.34 ± 0.02 | 2.1 ± 0.9 | 3.0 ± 0.5 | 8.2 ± 0.2 |
| 8 | 7.65 ± 0.05 | 58 ± 2 | 0.17 ± 0.01 | 9.6 ± 0.8 | 1.1 ± 0.1 | 2.3 ± 0.2 |
| 9 | 7.8 ± 0.1 | 64 ± 1 | 0.08 ± 0.01 | 3.3 ± 0.9 | 0.3 ± 0.1 | 0.47 ± 0.02 |

Regarding clays, even after the removal of carbonates, the concentration of the samples did not reveal any clays other than illite (902 cm$^{-1}$, 977 cm$^{-1}$, broad band 3000–3600 cm$^{-1}$; 3616 cm$^{-1}$) and kaolinite. (912 cm$^{-1}$; 1001 cm$^{-1}$, 1026 cm$^{-1}$, 3618 cm$^{-1}$, 3650 cm$^{-1}$, 3689 cm$^{-1}$) (Figure 1). These clays represent the typical mineralogy of the region, as we will discuss later in the next section. As a result, these clays contribute to the low CEC values observed in the soils (Table 1). Moreover, in the presence of asbestos, bands at 755–790, 937–943 and 3675 (Figure 3), or 3688 cm$^{-1}$ for chrysotile, would appear. However, none of these bands were present in the FTIR spectra of the soils, even in the samples treated with HCl where clays and silicates were more concentrated.

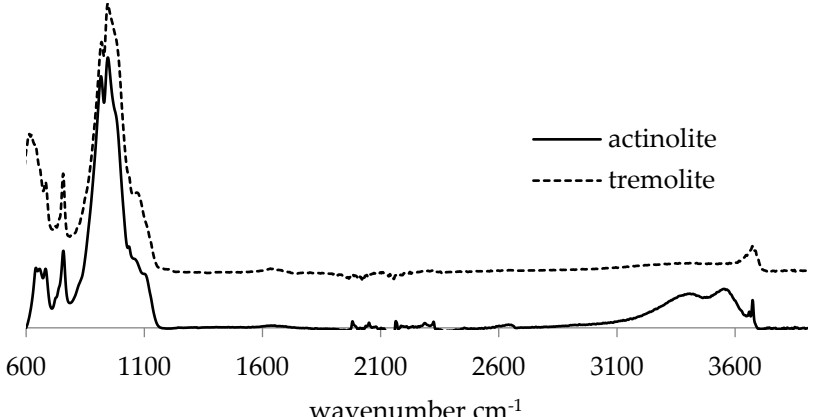

**Figure 3.** FTIR spectra of actinolite and tremolite. These spectra are used as standards. None of the bands observed in these spectra appeared in the soil spectra.

Significant variability in the levels of OC (Table 1) was observed with variations depending on the site and the year. This was not surprising, since some plots have been conditioned with plant debris. In general, the OC content was higher in 2019 than in 2020, although the OC in soils 4 and 7 was much greater in 2020 than in 2019.

### 3.2. Soil Elemental Composition

The elemental composition of soils is shown in Table 2. Soils were dominated by Ca, accordingly with high levels of calcite detected. Silicon was also present in high quantities, associated with the presence of illite and kaolinite clays. On the contrary, the Mg concentration was quite low as was expected from the lack of dolomite and Mg-bearing silicate bands in the FTIR spectra. It is important to highlight that heavy metals were not detected in significant quantities. However, Na and Cl levels were elevated in some samples, although

these concentrations were not high enough to result in high conductivity. The potential of heavy metal mobilization was studied by extraction with DTPA. The results of the extraction for the years 2019 and 2020 are shown in Tables 3 and 4, respectively. As it was expected, the mobility of metals was quite low due to the presence of lime and the high pH values in these soils. For example, only 0.005% of the total Al measured was solubilized, confirming that this element was primarily situated within clay structures, which are quite insoluble.

Elements that were most effectively extracted with DTPA included those for which the method was initially designed, such as Fe, Cu, Mn and Zn. In addition, notable extraction was observed for Pb, Al and Ni. Other potentially harmful elements appeared in low concentrations. As OC, the results were different in the two years of study. In this case, the levels of extracted metals, in general, were higher in 2020 than in 2019, especially for Mn.

**Table 2.** Elemental composition of the soils measured by FRX with La as a reference. Hg was measured independently. The analysis was made in 2019.

| Element g/kg | Soil 1 | Soil 2 | Soil 3 | Soil 4 | Soil 5 | Soil 6 | Soil 7 | Soil 8 | Soil 9 |
|---|---|---|---|---|---|---|---|---|---|
| Na | 29 | 1.7 | 3.1 | 0.8 | 1.0 | 1.8 | 0.8 | 0.8 | 1.3 |
| Mg | 13 | 9.4 | 18 | 9.1 | 14 | 13 | 6.8 | 6.7 | 14 |
| Al | 32 | 25 | 59 | 31 | 41 | 35 | 22 | 25 | 45 |
| Si | 100 | 74 | 160 | 84 | 110 | 100 | 58 | 72 | 130 |
| P | 0.5 | 0.4 | 1.1 | 1.0 | 1.7 | 1.0 | 1.6 | 0.6 | 0.7 |
| S | 1.4 | 2.5 | 1.9 | 2.5 | 1.3 | 1.4 | 2.9 | 2.2 | 2.0 |
| Cl | 34 | 0.5 | 1.5 | 0.5 | 0.1 | 0.5 | 0.9 | 0.3 | 0.3 |
| K | 14 | 9.0 | 22 | 12 | 16 | 13 | 8.6 | 9.1 | 16 |
| Ca | 340 | 340 | 440 | 220 | 250 | 260 | 220 | 200 | 440 |
| Ti | 2.9 | 2.3 | 4.5 | 2.5 | 3.5 | 3.0 | 2.1 | 2.7 | 3.8 |
| Cr | 0 | 0 | 0 | 0 | 0.3 | 0.1 | 0 | 0 | 0.2 |
| Mn | 0.2 | 0.2 | 0.5 | 0.4 | 0.5 | 0.4 | 0.3 | 0.2 | 0.4 |
| Fe | 19 | 15 | 34 | 21 | 27 | 21 | 16 | 18 | 27 |
| Cu | 0.11 | 0 | 0 | 0.07 | 0.07 | 0.07 | 0 | 0.08 | 0 |
| Zn | 0 | 0 | 0 | 0 | 0 | 0 | 0 | 0 | 0 |
| Br | 0.1 | 0 | 0 | 0.08 | 0 | 0 | 0 | 0.04 | 0 |
| Rb | 0.1 | 0.1 | 0.2 | 0.1 | 0.1 | 0.1 | 0.1 | 0.1 | 0.2 |
| Sr | 1.1 | 1.4 | 1.4 | 0.8 | 0.9 | 1.0 | 1.3 | 0.6 | 1.6 |
| Y | 0 | 0 | 0.04 | 0 | 0.03 | 0 | 0.01 | 0.02 | 0.03 |
| Zr | 0.4 | 0.0 | 0.2 | 0.2 | 0.0 | 0.3 | 0.1 | 0.1 | 0.3 |
| Ba | 0.0 | 0.3 | 0.4 | 0.3 | 0.3 | 0.2 | 0.1 | 0.3 | 0.4 |
| W | 0.4 | 0.3 | 0.5 | 0.3 | 0.3 | 0.2 | 0.6 | 0.2 | 0.4 |
| Co | 0 | 0 | 0.1 | 0 | 0.09 | 0 | 0 | 0 | 0 |
| Ni | 0 | 0 | 0 | 0 | 0 | 0 | 0 | 0.04 | 0 |
| Pb | 0 | 0 | 0 | 0 | 0 | 0 | 0 | 0.09 | 0 |
| Hg | 0.00002 | 0.00002 | 0.00002 | 0.00003 | 0.00002 | 0.00001 | 0.00003 | 0.00003 | 0.00001 |

**Table 3.** Soil metal and metalloid concentrations after DTPA extraction in 2019. Standard deviations are included.

| Element mg/kg | Soil 1 | Soil 2 | Soil 3 | Soil 4 | Soil 5 | Soil 6 | Soil 7 | Soil 8 | Soil 9 |
|---|---|---|---|---|---|---|---|---|---|
| Al | 0.50 ± 0.05 | 0.72 ± 0.02 | 1.1 ± 0.3 | 0.68 ± 0.01 | 1.2 ± 0.2 | 0.9 ± 0.1 | 3.0 ± 0.7 | 0.49 ± 0.06 | 0.4 ± 0.1 |
| Ti | 0.0177 ± 0.0004 | 0.020 ± 0.004 | 0.0283 ± 0.0008 | 0.027 ± 0.005 | 0.04 ± 0.01 | 0.028 ± 0.005 | 0.11 ± 0.03 | 0.012 ± 0.001 | 0.012 ± 0.007 |
| V | 0.095 ± 0.001 | 0.33 ± 0.01 | 0.088 ± 0.002 | 0.09 ± 0.01 | 0.16 ± 0.01 | 0.131 ± 0.006 | 0.101 ± 0.007 | 0.094 ± 0.001 | 0.112 ± 0.001 |
| Cr | 0.0036 ± 0.0005 | 0.0041 ± 0.0002 | 0.0049 ± 0.0006 | 0.0039 ± 0.0005 | 0.0046 ± 0.0009 | 0.0044 ± 0.0001 | 0.007 ± 0.001 | 0.0033 ± 0.0005 | 0.0033 ± 0.0004 |
| Mn | 4.3 ± 0.2 | 2.8 ± 0.2 | 5.7 ± 0.4 | 3.8 ± 0.9 | 4.7 ± 0.4 | 5.0 ± 0.2 | 3.9 ± 0.4 | 2.1 ± 0.1 | 1.7 ± 0.2 |
| Fe | 4.8 ± 0.4 | 6.0 ± 0.4 | 5.8 ± 0.5 | 4.9 ± 0.4 | 16 ± 1 | 8.1 ± 0.3 | 7.6 ± 0.8 | 3.6 ± 0.1 | 2.9 ± 0.1 |
| Co | 0.033 ± 0.002 | 0.0209 ± 0.0006 | 0.045 ± 0.003 | 0.03 ± 0.01 | 0.026 ± 0.004 | 0.0288 ± 0.0008 | 0.013 ± 0.001 | 0.0135 ± 0.0009 | 0.015 ± 0.002 |
| Ni | 0.1672 ± 0.0005 | 0.169 ± 0.005 | 0.42 ± 0.02 | 0.201 ± 0.004 | 0.38 ± 0.02 | 0.39 ± 0.03 | 0.26 ± 0.01 | 0.160 ± 0.006 | 0.14 ± 0.01 |
| Cu | 0.71 ± 0.06 | 0.70 ± 0.06 | 0.79 ± 0.01 | 0.68 ± 0.02 | 0.76 ± 0.05 | 0.55 ± 0.01 | 0.56 ± 0.04 | 0.40 ± 0.01 | 0.30 ± 0.03 |
| Zn | 4.8 ± 0.2 | 5.0 ± 0.4 | 4.1 ± 0.4 | 2.8 ± 0.3 | 3.6 ± 0.3 | 4.1 ± 0.3 | 9.6 ± 0.8 | 2.7 ± 0.2 | 0.52 ± 0.04 |
| As | 0.0145 ± 0.0004 | 0.0211 ± 0.0001 | 0.0142 ± 0.0002 | 0.0109 ± 0.0005 | 0.0112 ± 0.0008 | 0.0134 ± 0.0006 | 0.015 ± 0.001 | 0.0117 ± 0.0003 | 0.0047 ± 0.0001 |
| Mo | 0.013 ± 0.002 | 0.0205 ± 0.0003 | 0.020 ± 0.001 | 0.020 ± 0.001 | 0.00637 ± 0.00007 | 0.0069 ± 0.0001 | 0.023 ± 0.003 | 0.0065 ± 0.0002 | 0.013 ± 0.002 |
| Cd | 0.0182 ± 0.0004 | 0.00622 ± 0.00005 | 0.0101 ± 0.0003 | 0.00654 ± 0.00009 | 0.0240 ± 0.0008 | 0.0209 ± 0.0009 | 0.022 ± 0.001 | 0.0127 ± 0.0008 | 0.0026 ± 0.0001 |
| Sb | 0.0039 ± 0.0004 | 0.0031 ± 0.0005 | 0.00265 ± 0.00006 | 0.0044 ± 0.0004 | 0.00229 ± 0.00004 | 0.0023 ± 0.0001 | 0.0019 ± 0.0002 | 0.00282 ± 0.00006 | 0.0018 ± 0.00004 |
| Pb | 0.56 ± 0.02 | 1.43 ± 0.02 | 1.73 ± 0.03 | 4.266 ± 0.004 | 0.77 ± 0.02 | 2.5 ± 0.2 | 2.6 ± 0.1 | 3.91 ± 0.03 | 1.79 ± 0.09 |

**Table 4.** Soil metal and metalloid concentrations after DTPA extraction soils in 2020. Standard deviations are included.

| Element mg/kg | Soil 1 | Soil 2 | Soil 3 | Soil 4 | Soil 5 | Soil 6 | Soil 7 | Soil 8 | Soil 9 |
|---|---|---|---|---|---|---|---|---|---|
| Al | 0.097 ± 0.004 | 0.158 ± 0.006 | 0.18 ± 0.01 | 0.23 ± 0.02 | 0.13 ± 0.03 | 0.28 ± 0.02 | 0.46 ± 0.08 | 1.6 ± 0.2 | 0.134 ± 0.008 |
| Cr | 0.0046 ± 0.0008 | 0.0057 ± 0.0001 | 0.0061 ± 0.0001 | 0.0037 ± 0.0002 | 0.004 ± 0.001 | 0.0040 ± 0.0001 | 0.0059 ± 0.0004 | 0.008 | 0.003 ± 0.001 |
| Mn | 10 ± 2 | 15.5 ± 0.4 | 50 ± 2 | 70 ± 10 | 38 ± 4 | 72 ± 3 | 40 ± 1 | 84 ± 3 | 15 ± 1 |
| Fe | 1.75 ± 0.07 | 3.3 ± 0.1 | 4.3 ± 0.3 | 3.9 ± 0.2 | 25 ± 1 | 8.2 ± 0.3 | 6.7 ± 0.9 | 25 ± 2 | 2.84 ± 0.09 |
| Co | 0.17 ± 0.04 | 0.372 ± 0.009 | 0.86 ± 0.02 | 0.68 ± 0.08 | 0.4 ± 0.1 | 0.82 ± 0.04 | 0.42 ± 0.01 | 1.22 ± 0.04 | 0.2 ± 0.1 |
| Ni | 0.317 ± 0.008 | 0.333 ± 0.005 | 0.92 ± 0.01 | 0.51 ± 0.03 | 0.7 ± 0.2 | 0.56 ± 0.02 | 0.58 ± 0.01 | 0.73 ± 0.02 | 0.4 ± 0.2 |
| Cu | 2.41 ± 0.08 | 0.91 ± 0.08 | 1.8 ± 0.1 | 1.00 ± 0.06 | 1.3 ± 0.3 | 1.02 ± 0.05 | 1.38 ± 0.06 | 2.1 ± 0.1 | 0.9 ± 0.4 |
| Zn | 4.1 ± 0.2 | 36 ± 6 | 5.2 ± 0.4 | 5.3 ± 0.6 | 1.81 ± 0.02 | 2.8 ± 0.2 | 17 ± 1 | 11 ± 1 | 4.0 ± 0.4 |
| As | 0.032 ± 0.001 | 0.021 ± 0.006 | 0.025 ± 0.004 | 0.022 ± 0.002 | 0.011 ± 0.004 | 0.0184 ± 0.0005 | 0.043 ± 0.001 | 0.037 ± 0.003 | 0.005 ± 0.002 |
| Mo | 0.021 ± 0.002 | 0.052 ± 0.002 | 0.070 ± 0.003 | 0.048 ± 0.007 | 0.007 ± 0.002 | 0.0215 ± 0.0007 | 0.059 ± 0.002 | 0.013 ± 0.001 | 0.008 ± 0.004 |
| Cd | 0.075 ± 0.003 | 0.0076 ± 0.0004 | 0.030 ± 0.002 | 0.027 ± 0.002 | 0.03 ± 0.01 | 0.0261 ± 0.0009 | 0.032 ± 0.001 | 0.048 ± 0.003 | 0.007 ± 0.003 |
| Sn | 0.00041 ± 0.00008 | 0.00042 ± 0.00009 | 0.0004 ± 0.0001 | 0.0004 ± 0.0001 | 0.0003 ± 0.0001 | 0.00046 ± 0.00005 | 0.0013 ± 0.0002 | 0.0011 ± 0.0002 | 0.0011 ± 0.0005 |
| Sb | 0.00631 ± 0.0002 | 0.0051 ± 0.0002 | 0.0074 ± 0.0001 | 0.0059 ± 0.0004 | 0.005 ± 0.002 | 0.0088 ± 0.0006 | 0.00339 ± 0.00005 | 0.0197 ± 0.0005 | 0.004 ± 0.001 |
| Hg | nd | nd | nd | nd | nd | nd | 0.00002 ± 0.00004 | 0.00006 ± 0.00005 | nd |
| Pb | 4.0 ± 0.1 | 2.06 ± 0.08 | 7.16 ± 0.03 | 4.1 ± 0.1 | 2.08 ± 0.05 | 5.5 ± 0.2 | 3.27 ± 0.09 | 12.6 ± 0.3 | 2.4 ± 0.2 |

## 4. Discussion

The analysis of the soils developed in the present study did not reveal significant differences in their main properties when compared to other soils in the area [28,30]. They all were alkaline soils with varying organic matter content both in time and location and with clays with low absorption capacity. The natural fertility of Alicante soils is low. The carbonate content of the soil is always above 50%. Soil pH is not only high, but it is also due to the presence of carbonate. This means that, in addition to the immobilization of several nutrients by precipitation in the form of oxides, additional problems appear such as iron chlorosis, caused by the presence of bicarbonate [31]. The climate of the area corresponds to a semi-arid regime, with low soil organic matter content (around 1% in natural soils), and it is agriculture, with the addition of organic wastes, that raises the levels of soil organic matter. Organic matter content remains high and constant only in a few forested areas, but these areas are inland and not near the coast, as in our study. Thus, surprisingly, anthropized soils are usually higher in organic matter than natural soils. Organic matter provides nutrients and, what is more important for the area, is able to retain water for longer [32]. The fact that the organic matter content was variable in our soils implies that the fertility conditions were also variable and needed to be monitored. However, the agricultural productivity of the area is very high. This is not due to the soil but to the light and temperature. The region has high luminosity throughout the year as well as mild temperatures. The limiting factor of soil and, above all, water is solved by drip irrigation. Drip irrigation is the most common way to add water and fertilizers to crops in this area.

The mineralogical analysis by FTIR has been a valuable and straightforward technique for many soils in the region. In this work, we attempt to enhance the sensitivity of the technique by eliminating the main component (lime) in order to improve the detection of silicate fractions. However, the clays identified were consistent whether or not carbonates were present. The procedure of removing lime is long and tedious with no improvements in clay detection.

Both the clays detected, and the low levels of Mg obtained, in the elemental analysis suggest that, if asbestos was present, it was in low concentrations. Unfortunately, there is no unique analytical technique for understanding all the soil characteristics with absolute accuracy. This is especially true when analyzing solids. XRD detects the crystalline structure of minerals. The presence of quartz usually masks the signals of other minerals and, in addition, minerals must be in a crystalline form. Secondary ion mass spectrometry, laser microprobe mass spectrometry (LMMS), electron probe X-ray microanalysis (EPXMA) and X-ray photoelectron spectroscopy (XPS) have also been used for asbestos detection [33]; FTIR detects bond vibrations, so minerals may be present in amorphous form [34]; these authors considered "FTIR spectroscopy the most informative single technique not only for clay mineral composition and structure but also for interactions of the clay minerals with inorganic or organic compounds". Clays present characteristic O–H bands in the region between 3000 and 4000 cm$^{-1}$ [35], whose detection can be improved by calculating the second derivative of the spectrum [30]. In addition to doing that, we concentrated the clays by eliminating the carbonate phase and used a complementary technique such as elemental analysis. In spite of this fact, it is possible that asbestos particles were still masked among other clays. Studies have been completed on dispersing asbestos particles in talc and applying various techniques, especially microscopic and data analysis techniques [36,37]. In this way, very low asbestos detection limits can be achieved. However, these systems were relatively simple (lab blends of clays), and their success is not guaranteed in a medium as complex as soil. However, this question should be addressed in subsequent studies. Nevertheless, our study provided a better understanding of the soil's mineralogy, which is a permanent characteristic and offers insights into their effective management. Asbestos fibers are easily retained in soil pores and interact with the negatively charged surfaces of the soil. However, high pH values and fulvic acids can increase fiber mobility [38]. These experiments were carried out in columns of quartz sand and soil in the laboratory in the presence of chrysotile. It was found that by adding fulvic (FA), humic (HA) acids

or natural organic matter (NOM) to the columns, the transport of asbestos toward the bottom of the column was 10.4%, 4.4% and 0.4% for AF, HA and NOM, respectively. The authors attributed this effect to the dispersion of asbestos fibers by FA in the pore water, reducing the interaction with soil grains. Organic matter (OM) altered the surface charge of the chrysotile fibers from net positive to net negative, modifying their interaction with the soil. Application of compost to soil could lead to the movement of asbestos fibers to surface waters through shallow groundwater. However, it is necessary to take these experiments to the field in order to corroborate the laboratory data. It is possible that the high pH values of the soil and the presence of OC may have displaced asbestos to deeper layers if this material had ever been present. This could cause problems in aquifers but not in agricultural activities. While dust may be a concern, asbestos would be absent from the surface layers. Therefore, the possible risks associated with cultivating this type of soil may come from the presence of heavy and toxic metals and metalloids. The negative effects of heavy metals and metalloids on the human body have been widely described [39]. Arsenic causes malfunction of cellular respiration, cellular enzymes and mitosis. Lead affects the production of reactive oxygen species (ROS). When the concentration of ROS is high, structural damage occurs in cells, nucleic acids, membranes and lipids of any living being. Mercury mainly affects the brain but also the kidneys, muscles and nervous system. Like Hg, Cd can be absorbed and accumulated throughout life. In the same way, it remains in the soil for a long time, where it is absorbed by plants, thus entering the food web. Cadmium affects enzyme systems and oxidative stress [40]. Heavy metals cause severe problems not only to humans but also to the soil and plants. In a recent review, Alengebawy et al. [41] pointed out different effects of heavy metals on the soil and plant. Cadmium retains organic matter and can modify the physical and chemical properties of the soil. It also reduces the populations of microorganisms, root length, biomass, seed germination and the absorption and circulation of nutrients through the stem. On the other hand, Pb causes damage to DNA, atrophy and a decrease in the content of chlorophylls and proteins. The roots and stems are deformed in the presence of Cu, and it also affects the soil microbial populations. Modifications in enzymatic activity have also been described due to the effect of Zn, as well as chlorosis. Corn, wheat, barley, cauliflower, citrus fruits and vegetables can suffer necrosis and chlorosis due to the soil Cr content [42].

However, we did not observe levels significantly different from those of forest soils within the same area, although the mobilization of these metals could be a potential problem.

The DTPA metal extraction method is capable of solubilizing a wide range of elements and can be used for pollution tests [43]. It is noteworthy that some elements that were not detected by FRX were measured in DTPA extracts. This discrepancy can be attributed to several reasons. Firstly, because the detection limits are different in both methods. Secondly, it could be due to a concentration effect since some elements such as Si or carbonates are not extracted by this method. Additionally, some of the metals could be associated with organic forms, which could enhance their solubility [44]. The DTPA test was specifically developed to measure Fe, Cu, Mn and Zn available for plant uptake [29]. That is, the test established the minimum concentrations of these elements that must be required in the soil for optimal plant nutrition, rather than establishing maximum levels to prevent soil pollution. So, other data sources must be used in pollution studies. Table 5 presents several minimum and maximum levels of metals and metalloids collected from various soils, which are then compared to the soils studied in this research. Acceptable values for good plant nutrition are considered values of up to 10 mg/kg of Fe, Cu or Mn.



**Table 5.** Range of concentrations of elements extracted with DTPA in different types of soil. Data from this study are included.

| As mg/kg | Cd mg/kg | Co mg/kg | Cu mg/kg | Fe mg/kg | Mn mg/kg | Ni mg/kg | Pb mg/kg | Sb mg/kg | Zn mg/kg | References |
|---|---|---|---|---|---|---|---|---|---|---|
| | 0.01–0.39 | 0.01–0.47 | 0.60–28.4 | 8.29–97.7 | 14.19–89.1 | 0.47–15.2 | 0.81–17.7 | | 0.62–14.0 | Soils surrounding a steel production plant [45] |
| | | | | | | >1–143.0 | | | | Serpentine soils [46] |
| | | | | | | | 10–680 | | | Pb mine [47] |
| 5.0–15 | 34–37 | | | | | | 25–446 | 8.0–57 | 311–1375 | Mine soils [48] |
| | 0.1–6.9 | | 1.4–56 | | 3.5–87 | | 7.8–29.3 | | 9.6–362 | Soils amended with dredge sediments [49] |
| | | | | 1.8–304 | 1.8–115.5 | | | | 0.45–7.7 | Ebro valley soils [50] |
| | 0.05–0.09 | 0.1–0.17 | 4.5–10.23 | | | 0.9–1.09 | 1.99–2.98 | | | Paddy soils Egypt [43] |
| | 0.4–1.9 | | 2.4–23.2 | | | | 9.3–169 | | 40.8–455 | Soils polluted by metallurgic industry [51] |
| 0.0047–0.021 | 0.0026–0.024 | 0.013–0.045 | 0.3–0.79 | 2.9–16 | 1.7–5.7 | 0.14–0.42 | 0.56–4.2 | 0.002–0.004 | 0.52–9.6 | This work 2019 |
| 0.005–0.043 | 0.007–0.043 | 0.17–1.22 | 0.9–2.4 | 1.75–25 | 10.0–84 | 0.32–0.92 | 2.06–12.6 | 0.0034–0.02 | 1.81–36 | This work 2020 |

This increase in metal mobility during 2020 was not significantly associated with OC, despite the role of organic compounds in metal complexation. Interestingly, the soils were slightly richer in OC in 2019 than in 2020, although soils 4 and 8 reached very high levels of OC in 2020 (Table 1). Soils 8 and 9 also improved their OC concentration but to a lesser extent. These variations could be attributed to meteorological and soil management factors. In 2019, the year was particularly humid, reaching an average annual precipitation value of 647.7 mm and an average temperature of 18.4 °C [52,53]. In 2020, the average annual precipitation was 225.3 mm and the average annual temperature was 18.8 °C. That is, 2020 was drier and warmer than 2019. The increased moisture in 2019 might have facilitated better vegetation growth, contributing to the higher OC levels in the soils. Soils 4 and 8 are those that receive pruning remnants, and it is possible that the enhanced plant development in 2019 resulted in more pruning remains being added to these soils in 2020. In contrast, in the other soils where the vegetation growth was more constrained, fewer remains were added, leading to lower OC concentration in the soil.

Soil moisture could also be the cause of differences in metal extraction with DTPA. When soils are overdried in the laboratory for metal extraction, higher values of Cd and Zn are obtained [54]. Zinc tends to be immobilized in wet soils, while its extractability increases under oxidizing conditions [55]. Similar results were obtained for Fe as DTPA-extractable Fe was lower when moist soil samples were used compared to air-dry soil samples [56]. The above-average rainfall in 2019 may have contributed to the leaching of dissolved materials and soil fine particles, reducing elemental availability. On the other hand, the favorable natural vegetation development might have led to the elemental immobilization within living organisms. It has been recommended to conduct a thorough soil study before it is used in urban gardens. This can be especially important when these orchards are used as a hobby and the farmers are amateurs with little knowledge of soil management [14]. However, our findings highlight the necessity of ongoing annual risk monitoring, as numerous factors can influence metal availability. New risks may also appear, such as nanoparticles or microplastics [14], which can accumulate in the soil due to new industrial activities or even be deposited by wind or water. For these reasons, it is necessary to develop new rapid analysis tools that allow us to detect these contaminants quickly.

## 5. Conclusions

To our knowledge, this is the first time that the presence of asbestos in urban soils has been taken into consideration, even though with the analysis techniques used no problems have been detected in this regard. Despite the demonstrated advantages of expanding urban green areas from both environmental and social points of view, we should not forget the materials on which urban soils are developed. Hence, it is crucial to establish annual plans for monitoring variations in the availability of heavy metals. This becomes particularly relevant when plants are for human consumption. Consequently, it is also necessary to control the vegetables that grow in these soils. In the event of potential issues, it may be more prudent to use the soil for gardening instead of edible crops. Increasing the vegetation of cities, creating biodiversity spaces that allow for improved health and socialization, reducing the temperature of cities and obtaining safe vegetables cannot be completed without knowing and monitoring the soils where these activities are undertaken.

**Author Contributions:** Conceptualization, J.D.J. and A.S.-S.; methodology, M.C.; formal analysis, B.F.-G. and M.C.; investigation, B.F.-G., J.D.J., A.S.-S. and M.C.; writing—J.D.J. All authors have read and agreed to the published version of the manuscript.

**Funding:** This research received no external funding.

**Institutional Review Board Statement:** Not applicable.

**Informed Consent Statement:** Not applicable.

**Data Availability Statement:** https://rua.ua.es/dspace/ (accessed on 14 May 2023).

**Conflicts of Interest:** The authors declare no conflict of interest.

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
