# Peer review of "Characterization of Technosols for Urban Agriculture"

_sustainability, doi:10.3390/su152215769_

Round 1

Reviewer 1 Report

Comments and Suggestions for Authors

The submitted paper falls under the scope of the Journal of Sustainability, and I found it to be an interesting topic ‘’Characterization of Technosols for Urban Agriculture”’ and a relatively well-written paper.

 Still some issues are found in the manuscript which need to be address.

The abstract section is writhen good, but results are not well defining please improve it.  However, in abstract and methodology mention nine soils from the surroundings of Alicante area. So these are different area (location), different field or different soil series please add this information in method section. If the area is same soils collected from different points of the same area?? Write it.

 Please add two more keywords in the manuscript.

Introduction section

Lines 46 and 47 first write the full name of the element, then use abbreviations such as Cu, Mg, and Pb.

Line 98: All measurements were made in triplicate? Or, from one location, how many soil samples.

Line 71: Please modify the sentence and add (salinity, alkalinity) and a new citation to make the sentence complete. https://doi.org/10.3390/agronomy9120869

I think organic matter is a good characteristic of soil; it cannot affect the contamination of soil.

Soil characteristics such as salinity, alkalinity (soil pH > 7), and heavy metal competition affect the mobility and persistence of contaminants in soil.

Line 88 adds Alicante, Southeast Spain.

Line 98, All measurements were made in triplicate. Or from one location, how many soil samples were collected?

Comments on the Quality of English Language

Minor English needs improvement.

Author Response

Thank you so much for your comments. 

They have definitely contributed to the improvement of the article. We hope to have satisfactorily answered all questions.

The modified paper is included

The submitted paper falls under the scope of the Journal of Sustainability, and I found it to be an interesting topic ‘’Characterization of Technosols for Urban Agriculture”’ and a relatively well-written paper.

 Still some issues are found in the manuscript which need to be address.

The abstract section is writhen good, but results are not well defining please improve it.  However, in abstract and methodology mention nine soils from the surroundings of Alicante area. So these are different area (location), different field or different soil series please add this information in method section. If the area is same soils collected from different points of the same area?? Write it.

We have included the sentence: There are two industrial areas surrounding the housing area. The closest buildings have been abandoned and the industrial area is moving further and further away from the population. Between these two industrial zones and in a more or less straight line is where the studied soils are located (Fig.1).

A map of the area with the situation of the nine soils was also included

We have also modified the abstract

Please add two more keywords in the manuscript.

We have included climate and pollution

Introduction section

Lines 46 and 47 first write the full name of the element, then use abbreviations such as Cu, Mg, and Pb.

Sorry,  we disagree, we are not talking about abbreviations but rather universally accepted chemical symbols. For instance, Fe is the symbol for iron in English, whose abbreviation could be IR, but it is also the symbol for "hierro" in Spanish, whose abbreviation would be HI.

These are the chemical symbols according to the International Union of Pure and Applied Chemistry (IUPAC).It is common language in science

Line 98: All measurements were made in triplicate? Or, from one location, how many soil samples.

We have changed the sentence as

All measurements were made in triplicate, except for calcimetry that was quadruplicated

Soil location is explained by the new Fig 1

Line 71: Please modify the sentence and add (salinity, alkalinity) and a new citation to make the sentence complete. https://doi.org/10.3390/agronomy9120869

The sentence

In arid and semi-arid regions, soil salinity and alkalinity impose serious restrictions on plant growth; salinity limits water uptake by plants and reduces germination.

has been added with the reference

I think organic matter is a good characteristic of soil; it cannot affect the contamination of soil.

Yes in the 99% of the situations, it is, but in some occasions because of  a special characteristic of the organic matter I could cause some damage, this is what we wanted to explain

Soil characteristics such as salinity, alkalinity (soil pH > 7), and heavy metal competition affect the mobility and persistence of contaminants in soil.

Line 88 adds Alicante, Southeast Spain.

done

Line 98, All measurements were made in triplicate. Or from one location, how many soil samples were collected?

A map and explanation about the sampling has been included

Reviewer 2 Report

Comments and Suggestions for Authors

The manuscript (sustainability-2689079) analysed soils from the region in Alicante, Spain, which developed over construction debris and found no harmful elements but detected high levels of certain metals. Variations in organic carbon and metal contents over two years were noted and were influenced by rainfall and natural vegetation. Despite the benefits of urban green spaces, regular monitoring of heavy metals in soils is essential, especially for lands used in vegetable cultivation, to ensure safety and to address contamination by repurposing the soil for nonedible plants.

The sections of the manuscript have been well written. However, the authors must follow the “instructions for the authors. The introduction is informative and clear; however, the hypothesis and objectives need to be clarified further. In the Materials and Methods section, I recommend that the authors separate each distinct technique by subtopic to clarify the steps. In addition, a section for the statistical analyses and software used, as well as their modelling, should be detailed.

What statistical analyses were used in the manuscript? What was the actual number of samples used? A map of the soil collection, as well as of the city, should be incorporated to reflect the sampling points and their special distribution. Why do the authors, in Figure 2, employ a reference instead of conducting the appropriate analysis, either with a solo or standard of this class of soil? Table 3 is somewhat confusing with regard to the sample number.

The discussion section is interesting, but should rely more on the objectives and hypothesis - which should be explicit at the end of the introduction - but needs better bibliographic referencing.

Keywords in alphabetical order

L50. What units? “as 1000 mg/”?

The captions of the tables and figures require further improvement. They should provide as much description as possible related to the information they want to convey effectively.

Table 1. Why is there a blue stain in frame 8? Why is there an excessive number of decimal places?

L99. All units should be separated: 2 mm; 5 cm; 50 mL. Review of the manuscript:

Units: cmol kg−1;

References should not be added in the results section;

The discussion should be better grounded with a theoretical reference as well as associate the presented hypotheses.

My questions:

Why did the authors choose 9 technosols? What are the criteria for this choice?

What are the implications of soils being alkaline and having variable organic matter content on the environment and agricultural productivity in the area?

How effective is the FTIR method for detecting the mineralogical composition of soils, and are there alternatives that could offer better sensitivity and accuracy?

How can the movement of asbestos to deeper layers impact aquifers, and what mitigation measures can be implemented to prevent potential contamination?

What are the potential risks associated with the presence of heavy and toxic metals and metalloids in the soil, and how can they be monitored and managed effectively?

How do the pH values and fulvic acids in the soil impact the mobility and concentration of asbestos fibres, and what preventive measures can be taken to mitigate potential risks?

Comments on the Quality of English Language

English requires grammar and verbosity correction.

Author Response

Thank you so much for your comments. They have definitely contributed to the improvement of the article. We hope to have satisfactorily answered all questions.

We attach the revised manuscript, including English language and your questions

The manuscript (sustainability-2689079) analysed soils from the region in Alicante, Spain, which developed over construction debris and found no harmful elements but detected high levels of certain metals. Variations in organic carbon and metal contents over two years were noted and were influenced by rainfall and natural vegetation. Despite the benefits of urban green spaces, regular monitoring of heavy metals in soils is essential, especially for lands used in vegetable cultivation, to ensure safety and to address contamination by repurposing the soil for nonedible plants.

The sections of the manuscript have been well written. However, the authors must follow the “instructions for the authors. The introduction is informative and clear; however, the hypothesis and objectives need to be clarified further.

A sentence has been included at the end of the paragraph developing the  hypothesis

In the Materials and Methods section, I recommend that the authors separate each distinct technique by subtopic to clarify the steps.

The section has been subdividing according to your suggestion

In addition, a section for the statistical analyses and software used, as well as their modelling, should be detailed.

We have included this section

What statistical analyses were used in the manuscript? What was the actual number of samples used?

Done in the previous question

A map of the soil collection, as well as of the city, should be incorporated to reflect the sampling points and their special distribution.

The map has been included

Why do the authors, in Figure 2, employ a reference instead of conducting the appropriate analysis, either with a solo or standard of this class of soil?

We are not sure to understand this question. Obviously all soils were analyzed by FTIR, however, standards are used to check the bands. In addition each FTIR spectra is a sort of finger print that helps to the mineral identification. We have expanded the figure caption to make it clearer.

Table 3 is somewhat confusing with regard to the sample number.

The word soil has been removed

The discussion section is interesting, but should rely more on the objectives and hypothesis - which should be explicit at the end of the introduction - but needs better bibliographic referencing.

Keywords in alphabetical order

done

L50. What units? “as 1000 mg/”?

done

The captions of the tables and figures require further improvement. They should provide as much description as possible related to the information they want to convey effectively.

done

Table 1. Why is there a blue stain in frame 8? Why is there an excessive number of decimal places?

We cannot see this blue stain, it is maybe because the uploading process that transformed the manuscript. Any way, this table has been substituted by the location map

L99. All units should be separated: 2 mm; 5 cm; 50 mL. Review of the manuscript:

It is maybe again a problem with the document transformation once it has been uploaded to the platform, because we have completely reviewed and we have not detected this problem

Units: cmol kg−1;

Corrected in Table 2

References should not be added in the results section;

All te references in this section have been removed

The discussion should be better grounded with a theoretical reference as well as associate the presented hypotheses.

My questions:

Why did the authors choose 9 technosols? What are the criteria for this choice?

Taking samples uniformly in the area gave that result

What are the implications of soils being alkaline and having variable organic matter content on the environment and agricultural productivity in the area?

We included the following sentence in the discussion section:

The natural fertility of Alicante soils is low. The carbonate content of the soil is always above 50%. Soil pH is not only high, but it is also due to the presence of carbonate. This means that in addition to the immobilization of several nutrients by precipitation in the form of oxides, additional problems appear such as iron chlorosis, caused by the presence of bicarbonate [31]. The climate of the area corresponds to a semi-arid regime, with low soil organic matter content (around 1% in natural soils) and it is agriculture, with the addition of organic wastes, that raises the levels of soil organic matter. Only in a few forested areas, organic matter content remains high and constant, but these areas are inland and not near the coast, as it was in our study. Thus, surprisingly, anthropized soils are usually higher in organic matter than natural soils. Organic matter provides nutrients and, what is more important for the area, it is able to retain water for longer [32]. The fact that the organic matter content was variable in our soils, implies that the fertility conditions were also variable and needed to be monitorized. However, the agricultural productivity of the area is very high. This is not due to the soil but to the light and temperature. The region has high luminosity throughout the year, as well as mild temperatures. The limiting factor of soil and, above all, water is solved by drip irrigation. Drip irrigation is the most common way to add water and fertilizers to crops in this area. 

How effective is the FTIR method for detecting the mineralogical composition of soils, and are there alternatives that could offer better sensitivity and accuracy?

The following sentence has been included

Unfortunately, there is no a unique analytical technique to understand all the soil characteristics with absolute accuracy. This is especially true analyzing solids. XRD detects the crystalline structure of minerals. The presence of quartz usually masks the signals of other minerals and, in addition, minerals must be in a crystalline form. Secondary ion mass spectrometry, laser microprobe mass spectrometry (LMMS), electron probe X-ray microanalysis (EPXMA), and X-ray photoelectron spectroscopy (XPS) have also been used for asbestos detection [33]; FTIR detects bond vibrations, so minerals may be present in amorphous form [34], these authors considered “FTIR spectroscopy the most informative single technique not only for clay mineral composition and structure but also for interactions of the clay minerals with inorganic or organic compounds”. Clays present characteristic O-H bands in the region between 3000 and 4000 cm-1 [35], whose detection can be improved by calculating the second derivative of the spectrum [30]. In addition to doing that, we concentrated the clays by eliminating the carbonate phase, and used a complementary technique such as elemental analysis. In spite of the fact, it is possible that asbestos particles were still masked among other clays. Studies have been done dispersing asbestos particles in talc and applying various techniques, especially microscopic and data analysis ([36,37]. In this way, very low asbestos detection limits can be achieved. However, these systems were relatively simple (lab blends of clays) and their success is not guaranteed in a medium as complex as soil. However, this question should be addressed in subsequent studies

How can the movement of asbestos to deeper layers impact aquifers, and what mitigation measures can be implemented to prevent potential contamination?

See the last question

What are the potential risks associated with the presence of heavy and toxic metals and metalloids in the soil, and how can they be monitored and managed effectively?

We have included the sentence

The negative effects of heavy metals and metalloids on the human body have been widely described [39]. Arsenic causes malfunction of cellular respiration, cellular enzymes and mitosis. Lead affects the production of reactive oxygen species (ROS). When the concentration of ROS is high, structural damage occurs in cells, nucleic acids, membranes and lipids of any live being. Mercury mainly affects the brain, but also the kidneys, muscles and nervous system. Like Hg, Cd can be absorbed and accumulated throughout life. In the same way, it remains in the soil for a long time, where it is absorbed by plants, thus entering the food web. Cadmium is affects enzyme systems and oxidative stress [40].  Heavy metals cause severe problems, not only to humans but also to the soil and plants. In a recent review Alengebawy, et al. [41] pointed out different effects of heavy metals on the soil and plant. Cadmium retains organic matter, and can modify the physical and chemical properties of the soil. It also reduces the populations of microorganisms, root length, biomass, seed germination and the absorption and circulation of nutrients through the stem. On the other hand, Pb causes damage to DNA, leave atrophy and a decrease in the content of chlorophylls and proteins. The roots and stem are deformed in the presence of Cu, and it also affects the soil microbial populations. Modifications in enzymatic activity have also been described due to the effect of Zn, as well as chlorosis. Corn, wheat, barley, cauliflower, citrus fruits and vegetables can suffer necrosis and chlorosis due to the soil Cr content [42].

How do the pH values and fulvic acids in the soil impact the mobility and concentration of asbestos fibres, and what preventive measures can be taken to mitigate potential risks?

A more detailled description of the experiment was included

These experiments were carried out in columns of quartz sand and soil in the laboratory in the presence of chrysotile. It was found that adding fulvic (FA), humic (HA) acids or natural organic matter (NOM) to the columns, the transport of asbestos towards the bottom of the column was 10.4 %, 4.4 %  and 0.4 % for AF, HA and NOM, respectively. The authors attributed this effect to the dispersion of asbestos fibers by FA in the pore water, reducing the interaction with soil grains. Organic matter OM altered the surface charge of the chrysotile fibers from net positive to net negative, modifying their interaction with the soil. Application of compost to soil could lead to the movement of asbestos fibers to surface waters through shallow groundwater. However, it is necessary to take these experiments to the field in order to corroborate the laboratory data. 

Round 2

Reviewer 1 Report

Comments and Suggestions for Authors

The authors incorporate all my comments and recommendations.

Reviewer 2 Report

Comments and Suggestions for Authors

The authors clarified the doubts and improved the manuscript.

Comments on the Quality of English Language

Minor changes are needed in grammar.